# Effects of the Long-Term Climate Change and Selective Discharge Schemes on the Thermal Stratification of a Large Deep Reservoir, Xin'anjiang Reservoir, China

Huiyun Li [1], Jia Lan [2], Boqiang Qin [1,\*], Liancong Luo [3,\*], Junliang Jin [4], Guangwei Zhu [1] and Zhixu Wu [2]

1 State Key Laboratory of Lake Science and Environment, Nanjing Institute of Geography and Limnology, Chinese Academy of Sciences, Nanjing 210008, China
2 Hangzhou Bureau of Ecology and Environment Chun'an Branch, Hangzhou 311700, China
3 Institute for Ecological Research and Pollution Control of Plateau Lakes, School of Ecology and Environ-Mental Science, Yunnan University, Kunming 650504, China
4 Yangtze Institute for Conservation and Development, Nanjing Hydraulic Research Institute, Nanjing 210098, China
\* Correspondence: qinbq@niglas.ac.cn (B.Q.); billluo@ynu.edu.cn (L.L.)

**Abstract:** The effects of global warming and precipitation changes on water temperature and thermocline parameters, such as thermocline depth, thickness, and strength, were assessed. A catchment model, coupled with a reservoir thermal model with meteorological input calculated by a downscaled general circulation model (GCM) projection under three representative concentration pathways (RCPs), was applied to the Xin'anjiang Reservoir, located in southeast China. The results indicate that water temperature in each layer increased (decreased) with the rise (decline) in air temperature, especially the surface water temperature. There was a significant negative (positive) correlation between thermocline depth (strength) and air temperature during the period of stratification weakness. The most sensitive phenomenon of water temperature-to-precipitation changes occurred in the middle layer (depth = 30 m). Additionally, the thermocline depth and thickness increased with decreases in hydraulic residence time, which were caused by precipitation increases. According to the simulation experiments driven by RCP outputs, mean water temperature in each water layer in the future (2096–2100) has a strong response to increases in air temperature, which is projected to increase by 0.11–0.62 °C for RCP2.6, 0.76–1.19 °C for RCP4.5, and 1.50–2.35 °C for RCP8.5, compared to the baseline (2012–2016). However, mean water temperature in each water layer from 2096 to 2100 underwent a slight decrease caused by precipitation changes, with a 0.03–0.25 °C decrease for RCP2.6, 0.07–0.40 °C for RCP4.5, and 0.04–0.29 °C for RCP8.5, compared to 2012–2016. The mean thermocline depth in the future (2096–2100) will be significantly decreased, while the mean thermocline thickness will be slightly increased. Over a multiyear timescale, the impacts of air temperature changes are stronger than those induced by precipitation variations. However, the effects of hydraulic residence time changes caused by precipitation changes (especially rainstorm) should be considered in the management of deep reservoirs.

**Keywords:** climate change; thermal regime; stratification; numerical simulation; multiple scenarios

## 1. Introduction

The thermal regime is the most basic physical feature in lake and reservoir systems, which plays a very important role in biological metabolism and material decomposition [1–3]. In other words, water temperature and its seasonal variations significantly affect the structure of the biological community and the productivity of aquatic ecosystems in lakes and reservoirs [4,5]. For deep lakes and reservoirs, the thermal stratification determined by the difference in vertical water temperature is also the main factor associated with a variety of physical and chemical processes (such as dissolved oxygen distributive profile, light

penetration underwater, nutrient exchange between epilimnion and hypolimnion, and the photosynthesis of phytoplankton). For example, the water density stratification formed by thermal stratification restrains the mixing of the water column, making the hypolimnion rich in nutrition but insufficiently light, while the epilimnion is sufficiently light but has limited nutrition [6,7].

A reservoir formed by artificial damming is an important engineering measure for water resource management, such as water supply, flood control, power generation, irrigation, etc. [8,9]. The construction of the reservoir changes the characteristics of the natural river course by increasing water depth, decreasing flow velocity, and changing hydraulic residence time [10,11]. Reservoirs share their characteristics with both rivers and lakes and have unique physical characteristics. Usually, reservoirs are river-like at the head where major tributaries enter and are more lake-like near the dam. The vertical stratification due to temperature differences is one of the most typical and common features in the lake-like area of reservoirs [12]. However, for different reservoirs, the characteristics of thermal stratification and seasonal variations have significant individual differences. This can be influenced by reservoir morphology (depth and size), solar radiation, light penetration (transparency and diffuse attenuation coefficient), climate factors (air temperature, precipitation, and wind), as well as the differences in artificial regulation modes between reservoirs [13–15]. Generally, regional climate determines the thermal stratification formation and weakness periods of reservoirs. For example, deep reservoirs in tropical areas are stratified for most of the year, and there is only a short mixing period in winter. In addition, hydrological conditions and human regulation measures will influence the stability and duration of thermal stratification structures in reservoirs.

Since the 20th century, with the global warming, the temperature of reservoirs around the world has increased by varying degrees, which has a complex and profound direct and indirect impact on the physical, chemical, and biological processes of reservoirs [16,17]. On the one hand, long-term slow temperature rises and short-term extreme heatwaves will prolong the thermal stratification time, decrease the depth of the mixing layer and the thermocline depth, and increase the thermal stability [18–21]. Meanwhile, the diffusion depth of dissolved oxygen and oxycline depth significantly decrease, which strengthens the anoxic and anaerobic environment at the hypolimnion and sediment–water interface of the reservoir [22,23]. Correspondingly, the release of phosphate from the sediment is enhanced, which further boosts algae growth and proliferation [24,25]. However, at the thermal stratification period, rainstorm runoff can increase the water temperature at the bottom of the reservoir, weaken the stability of the thermal structure, and induce mixing in early autumn [26,27].

Reservoirs are built for many different purposes, including flood control, water supply, power generation, and navigation. Therefore, in addition to climate change, the evolution of reservoir water environment is also disturbed by human activities such as the reservoir operation regulation [12]. The impact of climate change on the thermal characteristics of a reservoir and consequences for oxygen, nutrients, and phytoplankton are more complex than for a natural lake. However, few studies have quantified such impacts on reservoirs [28,29].

Therefore, the main aims of this study are to: (1) analyze the thermal stratification cycle characteristics of a large-deep reservoir located at the subtropical and monsoonal climate area, (2) examine the sensitivity of water temperature at different depths and thermal stratification to meteorological factors (air temperature and precipitation) and reservoir operation, and (3) address the possible changes in thermal structure and stratification of the reservoir under three representative concentration pathways (RCPs) by using the outputs from a Coupled Model Inter-comparison Project Phase 5 (CMIP5) GCM, namely, CSIRO-Mk3.6.0, to drive a hydrological model coupled with a hydrodynamics model.

## 2. Materials and Methods

### 2.1. Study Area

The Xin'anjiang Reservoir (Figure 1), located in southeast China, is an artificial, large (area 573 km²) and deep lake (mean depth 30 m and maximum depth around 100 m), which was built in 1959. It is a long, narrow river-type reservoir formed by the construction of the Xin'anjiang hydroelectric dam. The lake has a capacity of 178.4 × 10⁸ m³ when the water level is at its normal height of 108 m asl. The hydraulic residence time is approximately 2 years. The basin area of the Xin'anjiang Reservoir is 10,442 km², with 5 main rivers (Xin'anjiang River, Dongyuangang River, Wuqiangxi River, Fenglingang River, and Yunyuangang River) flowing into the lake. Approximately 70% of the total inflow comes from these 5 rivers and over 50% of the total inflow comes from the Xin'anjiang River which is the biggest inflow river of the lake. The Xin'anjiang Reservoir is an important strategic water source in the Yangtze River Delta in China, supplying drinking water for the tens of millions of residents downstream. Additionally, the Xin'anjiang Reservoir is a famous tourist resort due to its beautiful scenery and excellent water quality.

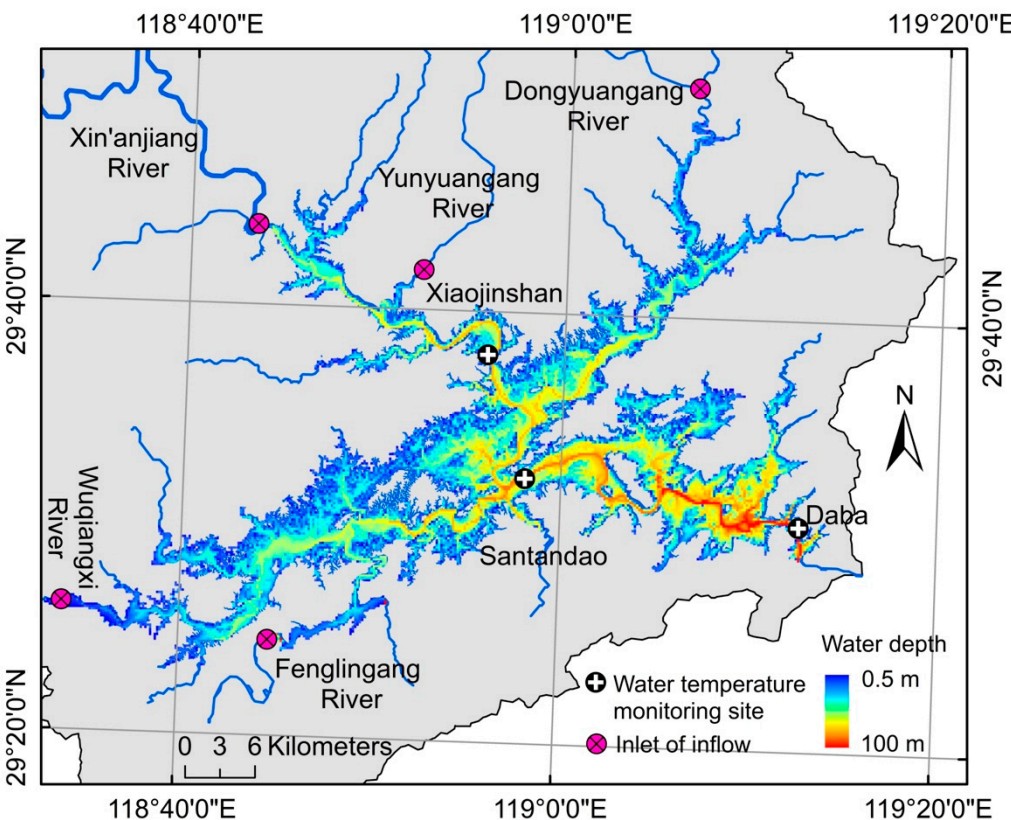

**Figure 1.** The location and water depth of the Xin'anjiang Reservoir, China.

### 2.2. Thermocline Detection

A common method (the gradient criterion method (GC)) is used to identify the thermocline in the Xin'anjiang Reservoir [30–32]. The mathematical expression of thermocline can be expressed as follows:

$$\partial T/\partial Z = (T_{i-1} - T_i)/(Z_i - Z_{i-1}), \qquad i = 2 \cdots K \tag{1}$$

where K is the total number of water layers from the surface to bottom of the lake and $Z_i$ and $T_i$ are the water depth and temperature of the ith layer, respectively. This is called normal distribution if the value of $\partial T/\partial Z$ is greater than zero; otherwise, it is called an inversion layer. It is necessary that the vertical temperature gradient should be larger than a certain fixed value, although there is no objective way to determine this criterion. It has

been reported that the criterion 0.2 °C/m is suitable when the water depth ≤200 m [33–35], and 0.05°C/m is adopted when the water depth >200 m [36,37]. The uniform criterion of 0.2 °C/m was chosen in this study based on experiences measuring the vertical distribution of temperature in the Xin'anjiang Reservoir.

Meanwhile, the thermal structure of the lake and reservoir can be also examined by calculating the distribution and variability of thermocline depth, thickness, and strength. Based on the equation mentioned above, thermocline depth is defined as the upper boundary depth of the thermocline layer. Thermocline thickness is the absolute difference between thermocline depth and the bottom depth of the thermocline. The average value of $\partial T/\partial Z$ between the upper and lower thermocline boundary is the strength of the thermocline.

### 2.3. Description of the Models

#### 2.3.1. DYRESM Model

The one-dimensional hydrodynamics model, called Dynamic Reservoir Simulation Model (DYRESM), for lakes and reservoirs was used in this study to simulate and predict the variation of water temperature, along with depth and time, for the Xin'anjiang Reservoir. The model was developed by the Centre for Water Research (CWR) at The University of Western Australia. The hydrodynamic component is process-based rather than empirical. The DYRESM model has shown good performance in predicting the vertical distribution of temperature, salinity, and density in lakes and reservoirs, satisfying the one-dimensional approximation [38–40]. More details on the theoretical background of the DYRESM can be found in the literature [41].

The DYRESM computer model does not require calibration because it relies on parameterizations derived from detailed process studies (both from the field and in the laboratory) [41]. The performance of the DYRESM model in simulating water temperature was evaluated using the root-mean-square-error (RMSE) and Pearson correlation coefficients (*r*) between estimated and observed values by the following equations [42,43]:

$$\text{RMSE} = \sqrt{\frac{1}{n}\sum_{i=1}^{n}\left[(S_i - \overline{S}) - (O_i - \overline{O})\right]^2} \qquad (2)$$

$$r = \frac{\sum_{i=1}^{n}(O_i - \overline{O})(S_i - \overline{S})}{\sqrt{\sum_{i=1}^{n}(O_i - \overline{O})^2}\sqrt{\sum_{i=1}^{n}(S_i - \overline{S})^2}} \qquad (3)$$

where n is the number of data records, $O_i$ and $S_i$ are the observed and simulated values, and $\overline{O}$ and $\overline{S}$ are the means of the observed and simulated values, respectively.

RMSE compares the actual difference between the estimated and the observed values term-by-term to provide information on the short-term performance of a model [44]. The smaller the value, the better the model's performance. Moreover, r reflects the linear correlation between two variables: O and S. The value of r is between −1 and 1. The greater the absolute value, the stronger the correlation.

#### 2.3.2. Rainfall–Runoff Model

The lumped daily rainfall–runoff model, called the Xin'anjiang model, developed by Zhao et al. (1980) [45], was used to estimate the streamflow entering the Xin'anjiang Reservoir. The model consists of four modules: a three-layer evapotranspiration module, a runoff generation module, a runoff separation module, and a runoff routing module. The schematic diagram of the Xin'anjiang model can be found in the study by Zhao et al. (1980, 1992) [45,46]. The parameters of the model were optimized using the generalized pattern search algorithm with linear inequality constraints in MATLAB. The Xin'anjiang model was calibrated by maximizing the Nash–Sutcliffe Efficiency (NSE) as follows [47,48]:

$$\text{NSE} = 1 - \frac{\sum_{i=1}^{n}(O_i - S_i)^2}{\sum_{i=1}^{n}(O_i - \overline{O})^2} \qquad (4)$$

NSE reflects the degree of agreement between simulated and observed values. NSE = 1.0 indicates perfect agreement, while NSE ≤ 1.0 indicates poor agreement. Meanwhile, the linear inequality constraint, called the water balance error (WBE), is used to compel the total simulated streamflow to within five percent of the total observed streamflow, as follows [49]:

$$\text{WBE} = 100 \left( \sum_{i=1}^{n} Q_{\text{sim,i}} - \sum_{i=1}^{n} Q_{\text{obs,i}} \right) / \sum_{i=1}^{n} Q_{\text{obs,i}} \tag{5}$$

*2.4. Data*

2.4.1. Field Data

To calibrate and validate the rainfall–runoff model, the observed streamflow data at the Tunxi hydrometric station (located in the upstream sub-catchment of the Xin'anjiang Reservoir Basin) for 2001–2018 were used. These observed streamflow data were sourced from the Hydrological Yearbook published by the Hydrological Bureau of Ministry of Water Resources of China. This time series was checked for outliers and errors for use in hydrological modeling.

To evaluate the performance of the DYRESM model in simulating water temperature for the Xin'anjiang Reservoir, a field investigation was carried out each month, from March 2012 to March 2014. The vertical distribution of water temperature was recorded using the XR-620 profiler (Richard Brancker Research Ltd., Ottawa, Ontario, Canada) at three sites: Xiaojinshan, Santandao, and Daba. When measuring the water temperature, the profiler was automatically dropped to the lake bottom by an electric winch at a speed of 0.1 m/s, with recording data every 2 s. Mean values of water temperature obtained from the three monitoring sites were used in this study to represent the average water temperature of the whole lake.

The water level data from March 2012 to December 2016, which were also used to evaluate the performance of the DYRESM model when simulating the water balance of the reservoir, were obtained from the local environmental monitoring station.

2.4.2. Observed Meteorological Data

To simulate the water temperature profile and the resultant stratification (or mixing) between layers, the DYRESM requires daily average input data for six meteorological variables, including air temperature (°C), short-wave radiation (W/m$^2$), cloud cover (fraction of whole sky), vapor pressure (hPa), wind speed (m/s), and rainfall (m). Daily meteorological data for 1959–2016 measured in the Tunxi and Chunan station (Figure 1) were downloaded from the China Meteorological Data Service Center (CMDC) (http://data.cma.cn/) (accessed on 1 May 2020) and preprocessed into the format required by the model.

In addition, air temperature and precipitation data from 1959 to 2005 were used to assess the performance of CMIP5 GCM outputs in the Xin'anjiang Reservoir Basin.

2.4.3. CMIP5 Data

Global climate models (GCMs) are important tools that can be used to assess the thermal stratification response of reservoirs and lakes to climate-driven forcing [50,51]. Four Coupled Model Inter-comparison Project Phase 5 (CMIP5) GCMs (https://esgf-node.llnl.gov/search/cmip5/) (accessed on 1 February 2020), including BCC-CSM1.1 (from China), CSIRO-Mk3.6.0 (from Australia), CCSM4 (from USA), and GISS-E2-R (from USA), were used to evaluate the applicability in the study area. The dataset most suitable for the Xin'anjiang Reservoir Basin will be selected to simulate the thermal stratification of the large deep reservoir in the future.

Each selected CMIP5 GCM dataset comprised three meteorological elements (daily precipitation, maximum temperature, and minimum temperature) for a historical period (1901–2005) and a future period (2006–2100). In data preprocessing, the GCMs data were downscaled into 0.5° × 0.5° by using a statistical downscaling method.

*2.5. Scenario Design*

2.5.1. Sensitivity Analysis Scenarios

Eight sensitivity analysis scenarios were designed to detect the influence of meteorological factors (air temperature and precipitation) on thermal structure. The control group simulation was from 2 March 2012 to 1 March 2013, 365 days in total. The DYRESM model was initialized with the vertical water temperature profile on 2 March 2012.

Four air-temperature scenarios that represent air temperature rising and decreasing were proposed to identify how air temperature changes affect water temperature, namely AIRT-4 (daily temperature is 4 °C lower than that of the control group), AIRT-2 (daily temperature is 2 °C lower than that of the control group), AIRT+2 (daily temperature is 2 °C higher than that of the control group), and AIRT+4 (daily temperature is 4 °C higher than that of the control group). All four scenarios assume that precipitation is the same as those in the control group.

Four precipitation-change scenarios (including OPER-20, OPER-10, OPER+10, and OPER+20) were carried out to analyze the effects of precipitation changes in the basin on reservoir water temperature. Generally, precipitation mainly affects reservoir thermal structure and external load by changing inflows, and then affects the chemical and biological processes of the reservoir. To ensure the optimum operation of the reservoir, these four scenarios assume that the water levels were the same as those in the control group. Accordingly, outflows and hydraulic residence time (reservoir volume divided by the mean annual discharge from the waterbody) are changes in all precipitation change scenarios [52]. Daily precipitation in the basin is 20% and 10% lower than that of the control group in OPER-20 and OPER-10 scenarios, and 10% and 20% higher than that of the control group in OPER+10 and OPER+20 scenarios.

2.5.2. RCP Scenarios

Three scenarios of representative concentration pathways (RCPs) were selected to assess how different emission scenarios impact water temperature in the Xin'anjiang Reservoir. These three scenarios include a stringent mitigation scenario (RCP2.6), an intermediate scenario (RCP4.5), and a very high greenhouse gas emissions scenario (RCP8.5), named according to their total radiative forcing in 2100 relative to pre-industrial values (+2.6, +4.5, and +8.5 $W/m^2$, respectively).

To address the effects of air temperature changes and precipitation changes in the future on vertical thermal structure, three groups of climate change scenarios (air temperature change, precipitation change, and interactive change) driven by RCP outputs were carried out. In the air-temperature-change scenarios (GP1), we used baseline precipitation records and air temperature outputs of CSIRO-Mk3.6.0 from 2096 to 2100 under RCP2.6, RCP4.5, and RCP8.5, respectively, to drive the hydrological model and DYRESM model. In the precipitation-change scenarios (GP2), we used baseline air temperature records and precipitation outputs of CSIRO-Mk3.6.0 from 2096 to 2100 (RCP2.6, RCP4.5, and RCP8.5) to drive the two models. In the interactive-change scenarios (GP3), the input files of air temperature and precipitation for the hydrological model and DYRESM model were all from CSIRO-Mk3.6.0 predictions (2096–2100, RCP2.6, RCP4.5, and RCP8.5).

## 3. Results

*3.1. Thermal Stratification*

3.1.1. Vertical Variation in Water Temperature

The annual changes in water temperature from March 2012 to March 2014 for each layer observed at the three sites in the Xin'anjiang Reservoir are shown in Figure 2. The arithmetic means of water temperature from the three monitoring sites represents the average of the whole reservoir.

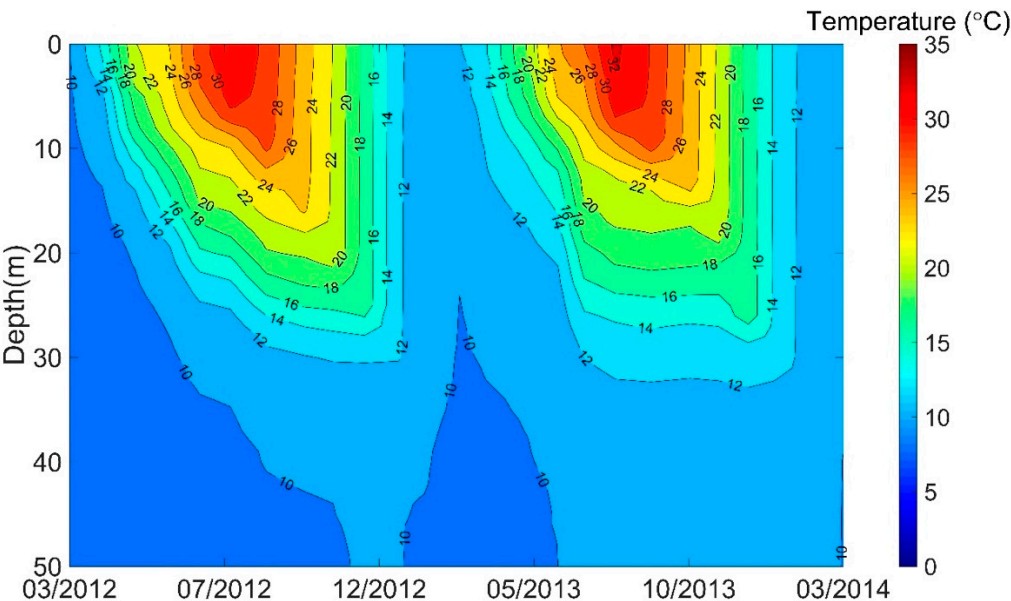

**Figure 2.** Depth–time diagram of isotherm (°C) in the Xin'anjiang Reservoir from March 2012 to March 2014.

In spring (March–May), the water temperature of the Xin'anjiang Reservoir began to vertically stratify, with the surface-layer (0–10 m) water temperature of 11.9–15.0 °C, middle-layer (10–30 m) water temperature of 9.7–11.9 °C, and bottom-layer (>30 m) water temperature of 9.4 °C. At the beginning of spring (March), the difference in water temperature between the surface layer and bottom layer of the reservoir was less than 3.0 °C. With the weather warming up, the surface waters of the reservoir heated more rapidly than the middle and bottom. The difference in water temperature in May between the surface layer and the bottom layer of the reservoir was greater than 10.0 °C.

In late summer (August), the surface-layer water temperature reached a peak value of 29.3–30.3 °C. Normally, the highest value of surface-layer water temperature lagged behind that of the air temperature by about one month, that is, the highest value of air temperature over the reservoir generally occurred in late July or early August. However, the seasonal variations in water temperature in the middle layer were inconsistent with that of the surface layer. The highest water temperature of the middle layer was observed in autumn (October), which ranged from 18.6 to 19.5 °C. For the bottom layer, there were no monthly and seasonal variations in water temperature, confirming that the water temperature of the bottom water layer was not affected by seasonal variations in air temperature. The difference in water temperature between the surface and the bottom was more than 15.0 °C in summer, with a maximum value of 19.7 °C in August. However, during the rest of the stratification period, from September to December, the difference in water temperature between the surface layer and the bottom layer gradually decreased.

From December to February of the next year, thermal stratification in the reservoir gradually disappeared. The water temperature of the surface layer decreased to 13.5–14.3 °C, as solar radiation was the weakest in a year. The difference in water temperature between the surface layer and the bottom layer was about 3.0 °C. There was a short mixing period in winter until the next thermal stratification formation.

3.1.2. The Formation and Development of the Thermocline

The monthly variations in the mean thermocline depth, thickness, and strength of the Xin'anjiang Reservoir are presented in Figure 3. For the whole reservoir, the thermocline was not obvious in March, but formed gradually in April. In late spring (May), the thermocline tended to be stable, with the thermocline depth of the reservoir being 3–5 m, the thermocline thickness being 12–17 m, and the thermocline strength being about 0.6–0.7.

From June to August, seasonal surface thermocline matured, the thermocline depth was relatively stable at 1–3 m, and the thermocline thickness gradually increased and reached the maximal value near 35 m. During this period, the mean thermocline strength of the Xin'anjiang Reservoir remained at the high record of 0.7. From the beginning of September to December, the thermocline depth significantly increased, and the thermocline thickness decreased month by month. No thermocline was monitored in March 2012, February–March 2013, and March 2014. Generally, the thermocline in the Xin'anjiang Reservoir began to form at the end of March or the beginning of April, gradually developed and matured from May to August, and became weaker and weaker from late September to December or January of the next year.

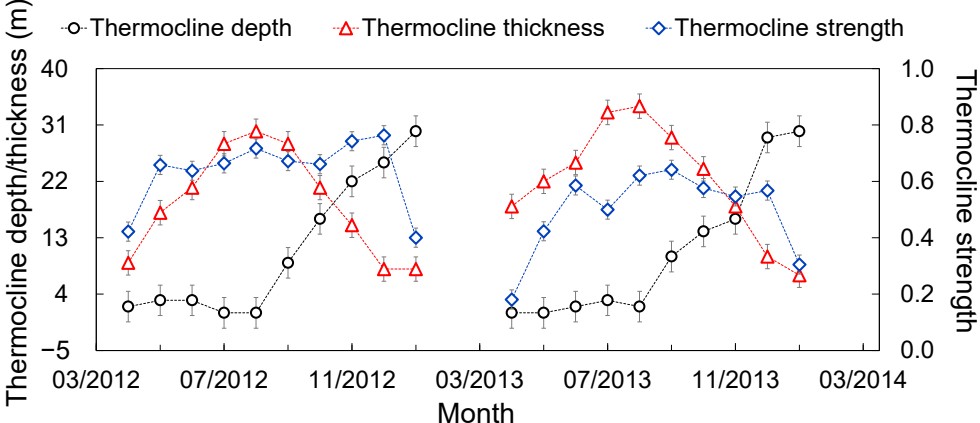

**Figure 3.** Monthly variations in thermocline depth, thickness, and strength from March 2012 to March 2014 in the Xin'anjiang Reservoir.

*3.2. Model Performance*

3.2.1. Performance of the Rainfall–Runoff Model

The results of the model calibration and validation for the Xin'anjiang model are shown in Figure 4. Two-thirds of the observed streamflow data (2001–2013) for the Xin'anjiang Reservoir Basin were used to calibrate the Xin'anjiang model and the calibrated parameter values were used to simulate streamflow for the remaining one-third period (2014–2018). In the timeline chart in Figure 4, the simulated streamflow (calibration and validation) agrees well with the available observed data. The NSE of streamflow and the total WBE for all the modelling runs are also shown in Figure 4. The line chart in Figure 4 shows that the rainfall–runoff model performed reasonably well in terms of model calibration, with high NSE values and low WBE values. The calibration NSE value was 0.94 for the Xin'anjiang model for the Xin'anjiang Reservoir Basin, while the total WBE in calibration was −4.16%. The observed and simulated streamflow over the non-calibration period were compared to determine the suitability of the model for this study. The results indicate that the rainfall–runoff models performed reasonably well in the validation period. The validation NSE value for the study basin was 0.90 for the Xin'anjiang model and the total WBE in validation was −13.59%. The total WBE in model validation was slightly higher than that in calibration but always below 15% for the Xin'anjiang Reservoir Basin, with an almost uniform plot of the underestimation and overestimation of flows. The calibrated model runs separately over five sub-catchments, and the runoff is then integrated.

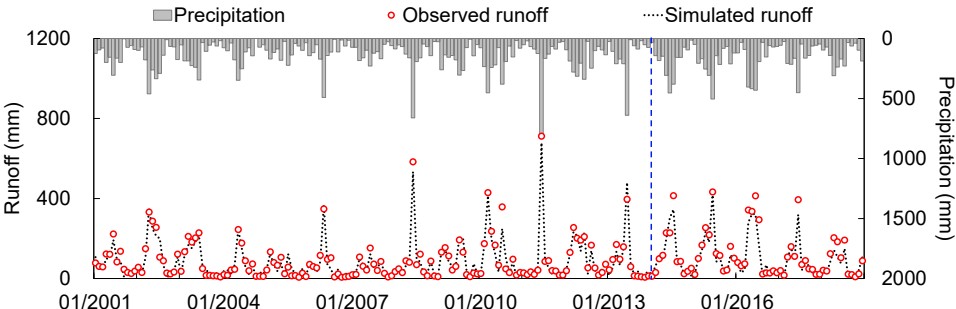

**Figure 4.** Comparison of modeled monthly streamflow obtained from the Xin'anjiang model and observed records of the Tunxi station in the Xin'anjiang Reservoir Basin.

3.2.2. Performance of DYRESM Model

The balance between precipitation, evaporation, water outflow, and inflow determined the water level in the Xin'anjiang Reservoir. After each simulation step, the model error was calculated, represented by the RMSE for water level. To gain further insight into the performance of the DYRESM model, the r between simulated water level and observed data was calculated. Figure 5 indicates that the DYRESM model could accurately reproduce water level during the whole simulation period (March 2012 to December 2016). The RMSE and r values for water level between the model output and the monitoring data were 0.13 and 0.999, respectively.

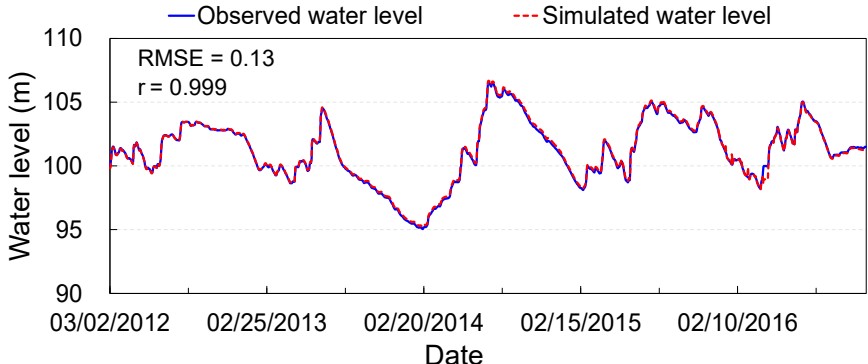

**Figure 5.** Comparison of modeled daily water level obtained from the DYRESM and observed records of the Xin'anjiang Reservoir.

As shown in Figure 6, there was a generally good agreement between the model output of water temperature and the monitoring data. The assigned values for the parameters used in the DYRESM are listed in Table 1. The heat budget reproduced the water temperature in all years of simulation quite well. For the epilimnion, the model produced the same seasonal dynamics as those that were measured. However, for the metalimnion and hypolimnion, it tended to over-estimate the measured values in the summer of 2013 and under-estimated the values in the subsequent winter. The RMSE values for water temperature in each layer between the model output and monitoring data were 0.96–1.78 °C. Meanwhile, the r values for water temperature in each layer between the modeling and monitoring values were 0.689–0.989. The results indicate that there is no evidence of systematic bias in the modeled water temperature of each layer in the model running period. The results suggest that the reservoir model used in this study is robust for an independent simulation period in different climate change scenarios.

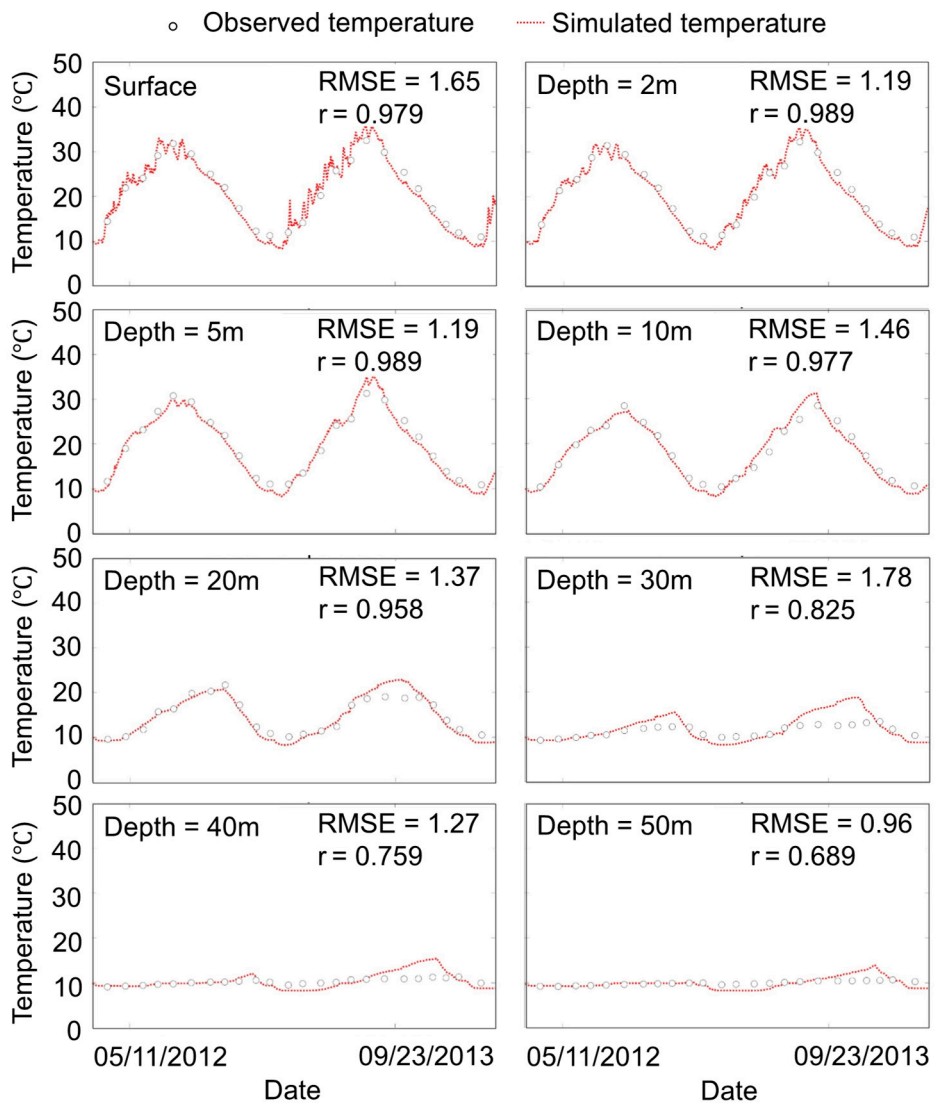

**Figure 6.** Comparison of simulated daily water temperature obtained from the DYRESM and observed data of the Xin'anjiang Reservoir from 2012 to 2014.

**Table 1.** Assigned values for parameters used in the DYRESM.

| Parameter Value | Description |
| --- | --- |
| $1.3 \times 10^{-3}$ | Bulk aerodynamic momentum transport coefficient |
| 0.088 | Mean albedo of water |
| 0.96 | Emissivity of a water surface |
| 6 | Critical wind speed |
| 0.012 | Bubbler entrainment coefficient |
| 0.083 | Buoyant plume entrainment coefficient |
| 0.08 | Shear production efficiency |
| 0.15 | Potential energy mixing efficiency |
| 0.15 | Wind stirring efficiency |
| $1.0 \times 10^{7}$ | Effective surface area coefficient |
| 0.15 | BBL detrainment diffusivity |
| 500 | Vertical mix coefficient |

### 3.2.3. Performance of CMIP5 Outputs

The Taylor diagram method proposed by Taylor was used to evaluate the applicability of datasets from CMIP5 GCMs to the simulation of historical meteorological indicators [53]. Two assessment criteria, RMSE and r, were adopted. A Taylor diagram can be plotted from

statistics of different series based on the relation of $RMSE^2 = \sigma_S{}^2 + \sigma_O{}^2 - 2\sigma_S\sigma_O r$, where $\sigma_O$ and $\sigma_S$ are standard deviations of observed values and standard deviations of simulated values, respectively. When the value of r is larger and the values of $\sigma_O$ and $\sigma_S$ are more approximate, the RMSE is smaller, which indicates that the model performs well. More detailed information about the Taylor diagram method can be found in the study by Taylor (2001) [53].

The performance of the four downscaled GCMs was assessed for each of the two meteorological predictors in the Xin'anjiang Reservoir Basin by plotting a Taylor diagram against CMDC data; see Figure 7. For monthly air temperature, all four CMIP5 models were suitable to simulate the historical data (1959–2005). CSIRO-Mk3.6.0 outperformed the other models, with the lowest RMSE of 1.865 and the highest r of 0.976 for monthly air temperature. The performance indexes of precipitation are generally qualified, but not as good as those of air temperature. For monthly precipitation, the RMSE between observed data and the four CMIP5 model outputs was 101.47–103.98 mm, while the r between the observed data and four CMIP5 model outputs was 0.483–0.561. Above all, the output from CSIRO-Mk3.6.0 was selected to estimate the effect of climate change between future and historical periods due to it having the best performance in climate simulations in the basin [54].

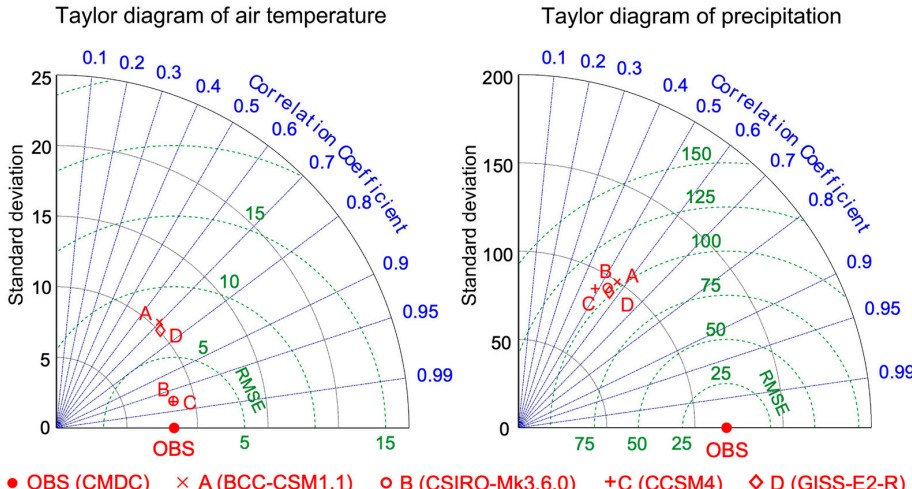

**Figure 7.** Taylor diagram of monthly air temperature and precipitation simulated by the four GCM models.

### 3.3. Sensitivity Analysis

3.3.1. The Sensitivity of Thermal Stratification under Air-Temperature-Change Scenarios

Before the simulation of future climate change effects, the sensitivity analysis of water temperature of the Xin'anjiang Reservoir was carried out to assess the impact of air temperature variations on thermal stratification by running a DYRESM model from 2 March 2012 to 1 March 2013. The sequences of input parameters for sensitivity studies on air temperature were generated by changing the absolute value of daily air temperature to historical meteorology records, then driving the model to obtain the responses of water-temperature profile time series. The results, shown in Figure 8a, illustrated that the water temperature from the surface to the bottom of the Xin'anjiang Reservoir was significantly sensitive to air temperature changes. Water temperature in each layer increases with increases in air temperature and decreases with decreases in air temperature. Figure 8a indicates that, as water depth increased, the variation ranges of water temperature under all four air-temperature-change scenarios markedly decreased. The maximum range of water temperature variations caused by air temperature changes was found in the surface layer and were likely to range from −2.10 °C to −1.90 °C under AIRT-4, −1.06 °C to −1.02 °C under AIRT-2, 0.92 °C to 1.09 °C under AIRT+2, and 1.85 °C to 2.19 °C under

AIRT+4. The minimum range of water temperature variations caused by air temperature changes was obtained in the bottom layer. When the air temperature increased (decreased) 1 °C, the bottom layer water temperature of the Xin'anjiang Reservoir increased by ~0.12 °C (decreased by ~0.2 °C).

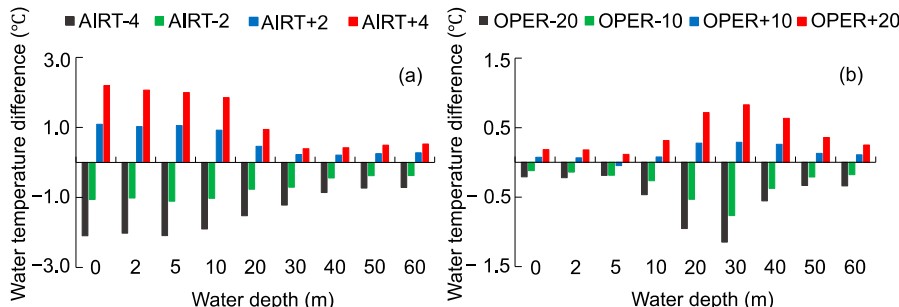

**Figure 8.** Differences in water temperature between each scenario and the baseline ((**a**) shows air-temperature-change scenarios and (**b**) shows precipitation changes with reservoir regulation scenarios).

The formation, maturation, and weakening processes of the thermocline simulated using the DYRESM model under the four scenarios of the air-temperature-change regime are shown in Figure 9 and Table 2. The results indicated that continuous warming would cause greater water-column stability and increase the duration of stratification compared to the absence of continuous warming. From October to December (the period of stratification weakness), the thermocline depth increased by 0.65–3.27 m under AIRT-4, 0.26–1.68 m under AIRT-2, decreased by 0.94–2.59 m under AIRT+2 and 1.29–5.09 m under AIRT+4. The duration of thermocline even extended to January of next year under two air-temperature-rising scenarios with the mean thermocline depths of ~43 m (AIRT+2) and ~44 m (AIRT+4), respectively. In contrast, there was no obvious regularity in the variation in thermocline depth under all air-temperature-change scenarios from March to September (the period of stratification formation and maturation). These results are consistent with the finding that a highly significant negative linear relationship existed between air temperature and thermocline depth in the period of stratification weakness but could not be found during the formation of stratification [32].

**Table 2.** Characterization of the stratification events under air-temperature-change scenarios and precipitation change with reservoir-regulation scenarios.

| Model Scenarios | Start of Seasonal Stratification | End of Seasonal Stratification | Duration of Thermocline (d) | Mean Surface Temperature (°C) | Mean Water Level (m) | Mean Inflow (m³/s) | Mean Outflow (m³/s) | Hydraulic Residence Time (d) |
|---|---|---|---|---|---|---|---|---|
| Baseline | 2012/3/14 | 2012/12/30 | 292 | 20.3 | 101.7 | 388.0 | 419.4 | 396 |
| AIRT-4 | 2012/3/23 | 2012/12/22 | 275 | 18.4 | 101.7 | 388.0 | 419.4 | 396 |
| AIRT-2 | 2012/3/22 | 2012/12/26 | 280 | 19.4 | 101.7 | 388.0 | 419.4 | 396 |
| AIRT+2 | 2012/3/14 | 2013/1/3 | 296 | 21.4 | 101.7 | 388.0 | 419.4 | 396 |
| AIRT+4 | 2012/3/13 | 2013/1/7 | 301 | 22.5 | 101.7 | 388.0 | 419.4 | 396 |
| OPER-20 | 2012/3/14 | 2013/1/5 | 298 | 20.3 | 101.7 | 278.7 | 296.0 | 561 |
| OPER-10 | 2012/3/14 | 2013/1/3 | 296 | 20.3 | 101.7 | 332.9 | 354.0 | 469 |
| OPER+10 | 2012/3/14 | 2012/12/25 | 287 | 20.3 | 101.7 | 443.6 | 495.0 | 336 |
| OPER+20 | 2012/3/14 | 2012/12/22 | 284 | 20.3 | 101.7 | 499.7 | 575.5 | 289 |

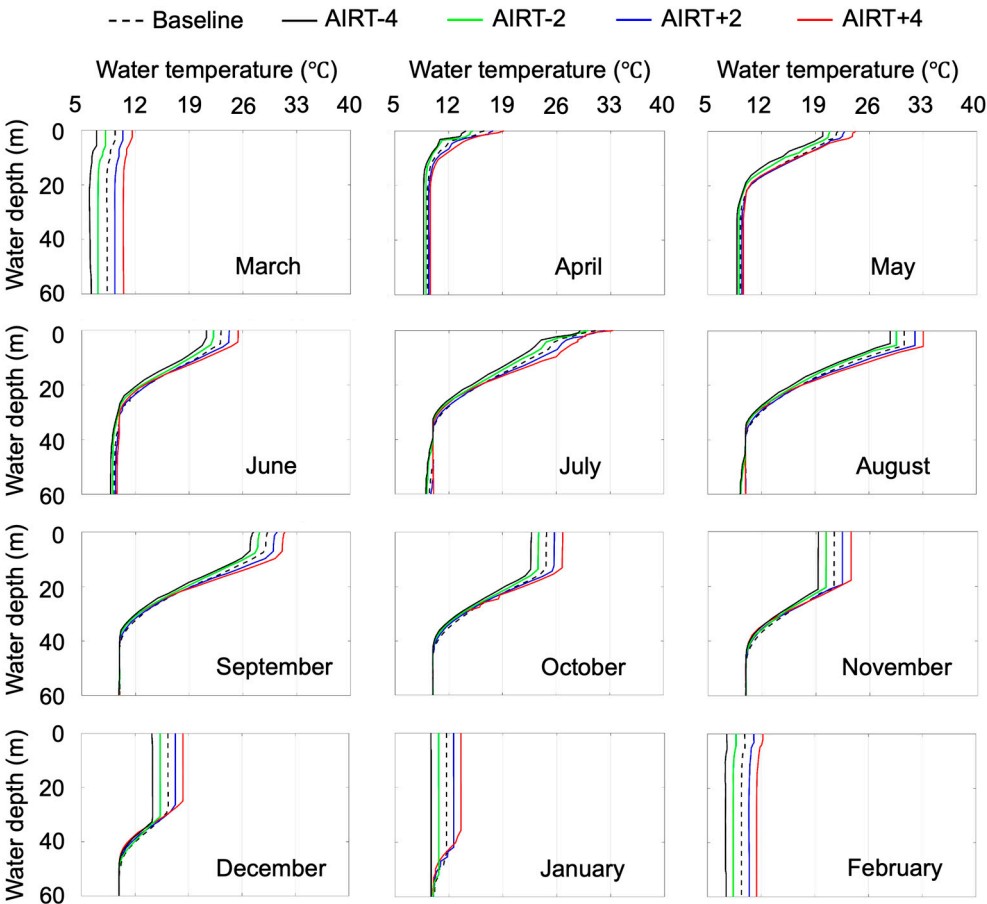

**Figure 9.** Simulated temperature profiles of the Xin'anjiang Reservoir under air-temperature-change scenarios (AIRT-4, AIRT-2, AIRT+2, and AIRT+4).

During October–December, mean thermocline thickness was significantly reduced by 2.92 m and 1.11 m under AIRT-4 and AIRT-2 and slightly increased by 0.74 m and 0.95 m under AIRT+2 and AIRT+4, respectively. However, in the other months, thermocline thickness changed under all four air-temperature-change scenarios but could not show obvious, regular changes.

Changes in thermocline strength caused by air temperature variations were significant and a clear pattern was discovered based on all the simulated outputs, including stratification formation, maturation, and weakness. As expected, higher air temperatures led to higher thermocline strength, while lower air temperatures led to lower thermocline strength. Mean thermocline strength of the whole sensitivity analysis period increased by 0.04 °C/m and 0.10 °C/m under AIRT+2 and AIRT+4 scenarios and decreased by −0.01 °C/m and −0.03 °C/m under AIRT-2 and AIRT-4, respectively.

### 3.3.2. The Sensitivity of Thermal Stratification under Precipitation Changes with Reservoir-Operation Scenarios

The outputs from the different model scenarios of precipitation change with reservoir regulation illustrated the reservoir thermal regime's response to variations in the climate and human activities (Figure 8b and Table 2). Figure 8b shows that changes in hydraulic residence time caused by outflow variations had slight effects on the surface and bottom temperature, but significant effects on middle-layer temperature, under four reservoir operation scenarios. Decreases in hydraulic residence time (OPER+10 and OPER+20) led to a higher water temperature, with a peak value of water temperature differences of +0.83 °C recorded in the middle layer compared to the baseline. In contrast, increases in hydraulic residence time (OPER-20 and OPER-10) resulted in a lower water temperature, with the

difference in water temperature ranging from −0.22 to −0.12 °C in the surface layer and −1.15 to −0.26 °C in the middle layer. There is a 70-m drop between the water surface of the reservoir and the water outlet elevation of the Xin'anjiang hydropower station located at the dam. The vertical mixing scheme before the dam would likely be significantly influenced by the increasing (decreasing) discharge from bottom of the dam. That is to say, surface water temperature, which has a significant positive linear relationship with air temperature, can dramatically affect the water temperature of the middle layer through this vertical mixing.

Monthly vertical water temperature profiles under different modeling scenarios of artificial regulation of reservoir, as shown in Figure 10, indicated that the effects of precipitation and hydrological conditions on thermal structure varied considerably with season. During the period of thermocline formation, March-July, the thermocline depth was rarely affected by whether precipitation and discharge increased or decreased. From the beginning of August to December, mean thermocline depth was significantly increased by 0.7 m under the OPER+10 scenario and 2.1 m under OPER+20 scenario and was significantly decreased by 1.1 m under the OPER-10 scenario and 1.8 m under OPER-20 scenario. Shortening the hydraulic residence time would dramatically increase thermocline depth in autumn and lead to early mixing in winter.

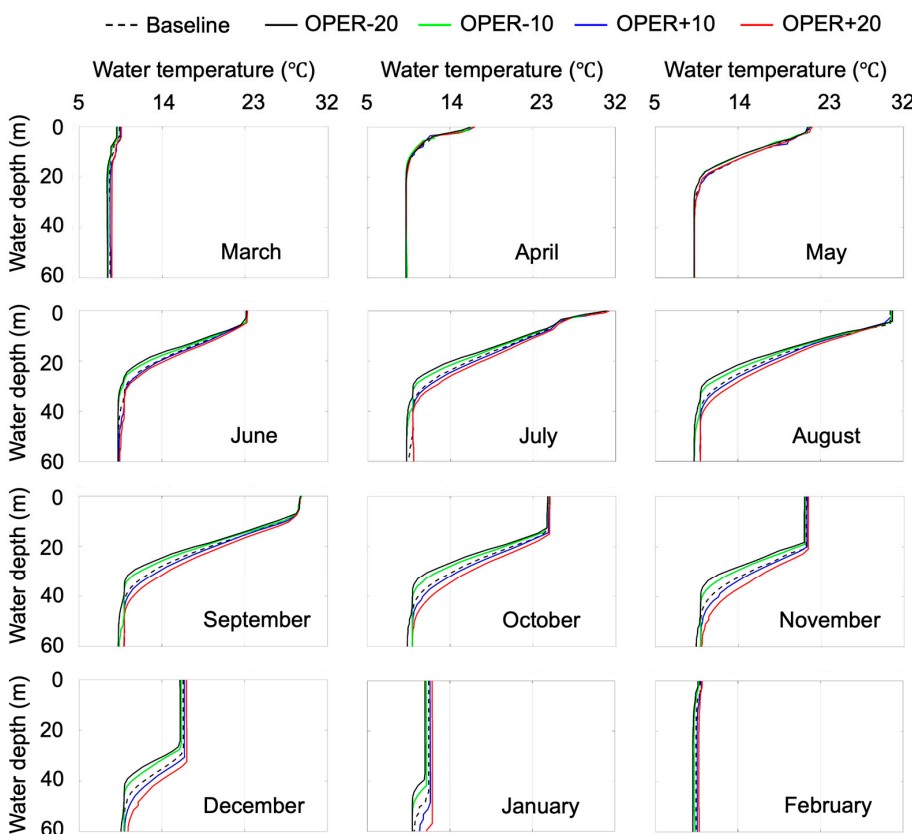

**Figure 10.** Simulated temperature profiles of the Xin'anjiang Reservoir under precipitation changes with reservoir regulation scenarios (OPER-20, OPER-10, OPER+10, and OPER+20).

Different from the change rules of thermocline depth, thermocline thickness was obviously affected by precipitation and hydrological conditions during matured and early weakness periods (June–October). When hydraulic residence time increased, the maximum difference in thermocline thickness was −5.16 m (recorded in August) under OPER-20 and −3.23 m (recorded in August) under OPER-10, respectively, compared to the baseline. However, when hydraulic residence time decreased, the maximum change in thermocline thickness was 2.94 m (recorded in October) under OPER+10 and 3.39 m (recorded in October) under OPER+20, respectively, relative to the baseline.

The monthly and seasonal variations in thermocline strength caused by precipitation and hydrological conditions were not uniform. The significant effects occurred from June to October. If the water level was kept the same as the baseline, the greater the rainfall and discharge (shorter hydraulic residence time) and the smaller the thermocline strength during this period. The peak value of increases in thermocline strength was 0.13 (in August) under OPER-20 and 0.09 (in July) under OPER-10, while the maximum decrease in the value of thermocline strength was 0.06 (in October) under OPER+10 and 0.07 (in October) under OPER+20.

### 3.4. Simulation of Thermal Structures in Future

The calibrated model parameters for both the baseline models (2012–2016) were used to simulate the inflow and water temperature of the Xin'anjiang Reservoir for the last 5 years of the 21st century (2096–2100). In the model simulations, the baseline calibrated parameters were used with the air temperature and precipitation time series for future investigations of the effects of climate change on the thermal structure between the two periods. Relative to the baseline, changes in the mean annual air temperature in the Xin'anjiang Reservoir basin over the period of 2096–2100 are projected to exceed 0.17 °C (0.94% of the baseline air temperature), 1.67 °C (9.14% of the baseline air temperature), and 3.94 °C (21.62% of the baseline air temperature) for RCP2.6, RCP4.5, and RCP8.5, respectively. The study region will continue to warm consistently with the global mean [55]. The decrease in annual precipitation in the study area over the prediction period is 14.6 mm (0.63% of the baseline precipitation) for RCP2.6, 120.45 mm (5.82% of the baseline precipitation) for RCP4.5, and 14.6 mm (0.63% of the baseline precipitation) for RCP8.5. Thus, in general, the climate in the prediction period (2096–2100) in the Xin'anjiang Reservoir Basin will be much warmer and dryer than that of the baseline (2012–2016). However, for long timescales, the increase in atmospheric $CO_2$ concentrations will make conditions around the Qiandao Lake much warmer and wetter.

The differences in water temperature were simulated in each layer between the baseline and future periods under these three groups of RCP scenarios (RCP2.6, RCP4.5, and RCP8.5). In the future, in 2096–2100, the mean annual water temperature in each layer shows a strong response to the increased air temperature, which is projected to increase by 0.5 (surface) to 5.9% (depth = 40 m) for RCP2.6 and by 10.1 (surface) to 19.1% (depth = 40 m) for RCP8.5. The maximum water temperature increase is at the depth of 40 m under RCP2.6 and the depth of 5 m under RCP8.5. This result is possibly because the CSIRO-Mk3.6.0 model sees the most pronounced warming in winter under RCP2.6. As air temperature clearly increases through the winter, the surface water becomes warmer and rapidly moves down in the short mixing period. However, under RCP8.5, the air temperature increases most dramatically in summer and the density gradient in the thermocline can act as a physical barrier that prevents vertical mixing between the epilimnion and hypolimnion during this period. The mean annual water temperature is slightly affected by precipitation changes, which decreased by 0.2 to 2.2% for RCP2.6 and by 0.2 to 2.4% for RCP8.5 relative to the baseline period. It is indicated that, at the end of the 21st century, the effect of air temperature changes on the vertical distribution of water temperature in the Xin'anjiang Reservoir is more significant than that of precipitation changes, as shown in Figure 11. As a good indicator of climate change in lakes and reservoirs, water temperature can reflect climatic forcing more immediately and sensitively than any other parameters [56,57].

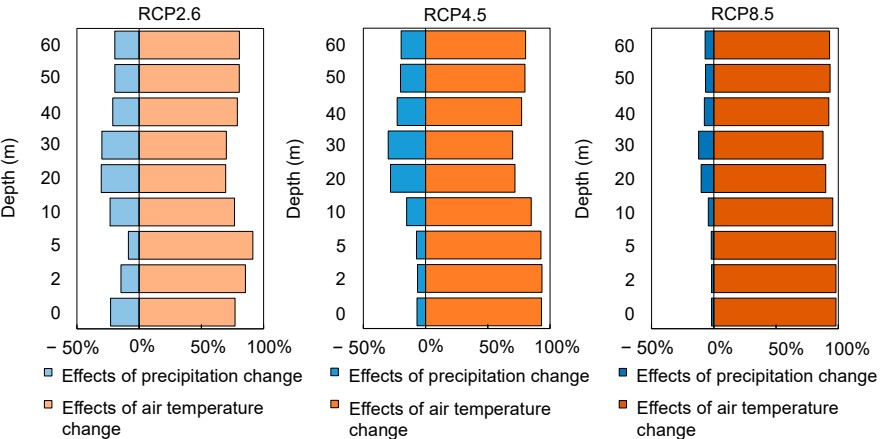

**Figure 11.** The influence of precipitation changes and air temperature changes on the water temperature of the Xin'anjiang Reservoir accounts for the proportion of the combined effect of these two meteorological factors under RCP2.6, RCP4.5, and RCP8.5 scenarios, respectively.

With global warming, the mean thermocline depth in the period of stratification weakness (October to December) from 2096 to 2100 in the Xin'anjiang Reservoir decreases by 0.52 m under RCP2.6 and 1.59 m under RCP8.5, as shown in Figure 12. During the same period, the mean thermocline thickness is slightly increased by 0.16 m and 0.84 m under RCP2.6 and RCP8.5, respectively. Increases in mean thermocline strength in the future are not obvious, at 0.02 °C/m under RCP8.5 (no increase under RCP2.6). The influence of climate change on the annual mean thermocline depth, thickness, and strength in the future is mainly governed by the increase in air temperature.

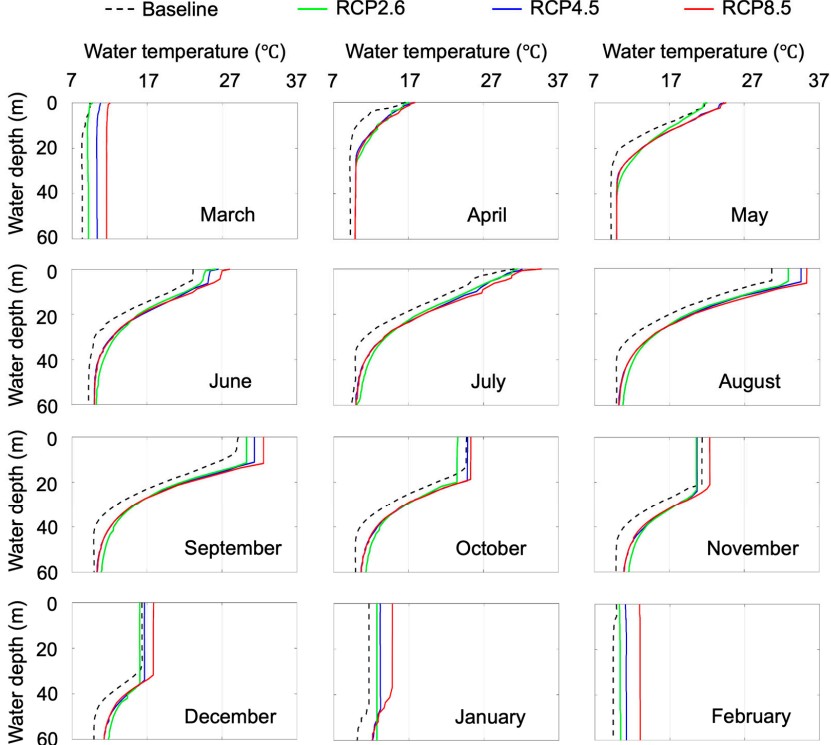

**Figure 12.** Simulated temperature profiles of the Xin'anjiang Reservoir under RCP2.6, RCP4.5, RCP8.5, and the baseline.

## 4. Discussion

### 4.1. Effects of Climate Change on the Thermal Stratification of Reservoirs

Thermal stratification forms the basic physical phenomena in the reservoir system, which is the main factor causing various physical and chemical processes [13,27,35]. It also plays a very important role in biological metabolism and material decomposition [58–61]. According to monitoring and modeling studies in recent years, we identified that the Xin'anjiang Reservoir was stratified for more than ten months in a year, with the water temperature gradient being negative during the stratification period.

Global warming causes different degrees of warming in global lakes and reservoirs, which has a complex and profound direct and indirect impact on the physical, chemical, and biological processes of lakes and reservoirs [16]. The phenomena of water temperature rise caused by climate change will change the thermal structure of the water body, affecting thermocline depth, thickness, and strength. Zhang et al. established a linear relationship between air temperature and water temperature of the Xin'anjiang Reservoir using the historical data and predicated that the increase in surface-water temperature would cause a decrease in the thermocline depth and increase in thermocline thickness and strength in the future [32]. Komatsu et al. simulated the influence of long-term climate change on the thermal stratification of the Shimajigawa reservoir in Japan and considered that the surface water temperature of the reservoir would increase by 3.8 °C in the last 10 years of this century [62]. The warming amplitude of the surface layer was greater than that of the bottom layer in summer, and the stratification time could be extended by 36 days at the most.

The sensitivity studies demonstrate that inflows of the reservoir are sensitive to long-term trends and the variability of precipitation in the basin. Since a primary function of reservoirs is to store water, the reservoir may stabilize the flow of water by regulating the downstream outflow. Thus, when the precipitation in the basin changes, the hydraulic residence time, which is defined as the reservoir volume divided by the outflow rate, is lengthened or shortened with changes in inflows and outflows. The shorter hydraulic residence time caused by increased daily precipitation in our study led to a higher thermocline depth and longer mixing period. Huang et al. monitored the effects of high inflows on the thermal regime of a deep, stratified reservoir in a temperate monsoon zone in Northwest China and considered that the increase in water temperature in the hypolimnion after the high volumes of inflow led to reservoir-mixing at the beginning of November [27].

However, the massive inflow induced by storm runoff might decrease thermocline depth because the increasing content of suspended solids in inflows can be caused by the rainstorms. After the storm runoff, the strong convective mixing effect increased the turbidity in the upper water of the reservoir. It has been demonstrated that there was a strong negative relationship between turbidity and thermocline depth in tropical and temperate lakes and reservoirs [63,64]. The reason for this phenomenon is that solar radiation penetrated the surface of the water decays more rapidly in high-turbidity conditions [56].

### 4.2. Effects of Thermal Stratification on Water Quality of Reservoirs against the Background of Global Warming

Thermal behaviors in reservoirs have a significant impact on the water quality process, especially for dissolved oxygen in water [65,66]. The strengthening of the water temperature stratification of reservoirs caused by global warming makes it difficult for the surface dissolved oxygen to penetrate into the bottom. The long-term anoxic and anaerobic conditions appear in the bottom of reservoirs due to the high concentration of organic matter and rapid oxygen consumption. For example, a high-frequency monitor of long thermal stratification events in Lake Müggelsee in Germany showed that hypolimnetic oxygen concentrations strongly decreased depending on the duration and intensity of stratification, and the low oxygen conditions could persist for up to a maximum of 9 days during the extreme summer in 2006 [6].

With global warming, the anaerobic conditions are more easily formed in the bottom layer of reservoirs, which promotes the release of nutrients from the sediment at the water–soil interface [67]. For a deep reservoir at the East Asian Monsoon, with air temperatures that gradually decrease in winter, the water temperature of the surface layer begins to decrease and the water density in surface layer increases accordingly. The upwelling and downwelling between the surface and bottom waters will take place as a result of the water density gradient being unstable. In this case, the algae in the surface layer will be brought to the bottom of the water, which can also replenish oxygen to the bottom, and the bottom nutrients can be brought to the surface with the rising water flow [61]. This phenomenon may accelerate reservoirs' eutrophication process and play a key role in algae growth.

Against the background of global warming, annual precipitation will slightly increase, and the frequency of extreme rainstorm events will be greatly increased in the future [68]. The extreme rainfall events caused by global warming will also have a great impact on water quality by affecting the thermal stratification of the reservoir. Intense rainfall–runoff is responsible for the short-term cyanobacterial blooms that occurred in many lakes or reservoirs [69–71]. The main result is that the water temperature of the rainstorm flood is higher than that of the reservoir bottom layer during the thermal stratification in summer. Meanwhile, a large amount of sediment with nutrients carried by the upstream flood will result in a significant increase in the density of inflows. Therefore, the high-density inflows will infiltrate the bottom of the reservoir to form the muddy water density current and increase the water temperature at the bottom of the reservoir. The stability of the thermal stratification structure will be weakened, and the reservoir water body will be induced to mix ahead of time. During the period of stratification weakness of the thermocline, nutrients such as nitrogen and phosphorus that accumulated at the bottom of the reservoir are transported to the upper water body, thus providing sufficient nutrients for the growth of algae in the coming year. This phenomenon has been reported in many lakes and reservoirs [27,72,73].

## 5. Conclusions

A rainfall–runoff model and a hydrodynamics model were coupled to investigate the potential effects of long-term climatic change and artificial operation on the thermal structure of a reservoir. The model validation indicates that this coupled model can be applied to the assessment of climatic change impacts on large-deep reservoirs in the East Asian monsoon regions. The main conclusions are summarized as follows:

(1) The water temperature in each layer was sensitive to changes in air temperature and showed a positive correlation with these changes. However, the influence of rising air temperature on thermal stratification parameters is inconsistent.

(2) The effects of precipitation variations in water temperature are more complicated than those of air temperature because of the reservoir operation involved. The middle layer (at the water depth of about 30 m) was the most sensitive water layer.

(3) With the global warming, the peak value of water temperature rises occurred at depth = 40 m under RCP2.6, but at depth = 5 m under RCP8.5. The mean thermocline depth in the future (2096–2100) will significantly decrease, while the mean thermocline thickness will slightly increase.

Our findings in this work indicate that air temperature is the main factor affecting the future thermal structure of the Xin'anjiang Reservoir; however, the effects of hydraulic residence time changes caused by inflow and outflow volume variations (especially during rainstorm period) should not be ignored.

**Author Contributions:** Conceptualization, B.Q. and L.L.; methodology, H.L.; software, H.L. and L.L.; validation, H.L., J.L., and J.J.; formal analysis, H.L. and J.L.; investigation, G.Z. and Z.W.; resources, J.L., G.Z. and J.J.; data curation, J.L.; writing—original draft preparation, H.L.; writing—review and editing, B.Q. and L.L.; visualization, H.L.; supervision, B.Q. All authors have read and agreed to the published version of the manuscript.

**Funding:** This research was funded by the National Natural Science Foundation of China (42171034 and 41830757), The National Key Research and Development Program of China (2022YFC3200134) and Yunnan University (C176220100043).

**Institutional Review Board Statement:** Not applicable.

**Informed Consent Statement:** Not applicable.

**Data Availability Statement:** The relevant data are available upon request to the corresponding authors.

**Conflicts of Interest:** The authors declare no conflict of interest.

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
