# Peer review of "Effects of the Long-Term Climate Change and Selective Discharge Schemes on the Thermal Stratification of a Large Deep Reservoir, Xin’anjiang Reservoir, China"

_water, doi:10.3390/w14203279_

Round 1

Reviewer 1 Report

The manuscript “Effects of the long-term climate change and selective discharge schemes on the thermal stratification of a large deep reservoir, Xin’anjiang Reservoir, China” is very interesting and some suggestions are as follow:

1-      The language needs revision.

2-      Work more with the cohesion of the work.

3-      What is the novelty of the research?

4-      Why this model of reservoir modeling?

5-      Why this region is important?

6-      Conclusion also needs revision and do not repeat the results again.

7-      Some suggested references:

-Downscaling the contribution to uncertainty in climate‐change assessments: representative concentration pathway (RCP) scenarios for the South Alborz Range, Iran, Meteorological Applications,2018.

-An ensemble forecast of semi‐arid rainfall using large‐scale climate predictors, Meteorological Applications,2017.

Reviewer 2 Report

The manuscript is very interesting,  however, the important factor influencing the temperature distribution in the water reservoir was not taken into account. It is about the inflow of groundwater to the reservoir, i.e. groundwater drainage.

It would be necessary to construct also a model of  groundwater flow. This would help determine the characteristics of groundwater and surface water exchange.

Possibly the authors could explain why they omitted it

Reviewer 3 Report

Thank you for providing the opportunity to review this manuscript which focuses on evaluating thermal stratification characteristics of a reservoir as a function of potential climate-related changes in temperature and precipitation. Overall, I felt this manuscript was well written, complete in design and informative. I don’t have any major concerns with the manuscript other than what I list below. The manuscript would benefit from a final review for technical writing. There were numerous instances of misused used tenses, e.g., are vs. were, is. vs. was. Also, line numbers were not provided.

Specific Points:

Abstract: Abstract is a bit long and should be streamlined. Also, the first sentence should be “global warming and precipitation changes on water temperature…”

Page 2, 3rd para: Say “Since the 20th century, the global reservoir…”

Page 3, 1st para.: Based on what is written, it is unclear how the CMIP5 GCM models will be used. What is meant by “employ”?

Page 4: Describe what the Chinese shelf is.

Page 7, section 3.1.1: Is what is described in this section observed or modeled data?

Figure 3: Describe what “Thermocline strength” means in the figure caption.

Figure 7: It appears the authors are testing the validity/accuracy of the CMIP5 data. This true? If so, provide justification.

Page 14, 1st para.: Say “lead to higher…”

Page 15: How was residence time calculated?

Page 15, 3.4: Explain what RCP2.6, RCP4.5 and RCP8.5 mean as far as temperature and precipitation change. How different are temperature and precip. for each of these?

Page 17: Say “Effects of climate warming on thermal…”

Page 17, 2nd para: What is meant by “water temperature rises phenomena….”?

Page 19, 1st para: What is meant by “extinction”?

Page 19, 2nd para: Say “For this research we used a coupled…”

Page 19, 3rd para: The first three sentences appear to be contradictory. Rephrasing needed.
